# Effects of *Spirulina platensis* and/or *Allium sativum* on Antioxidant Status, Immune Response, Gut Morphology, and Intestinal Lactobacilli and Coliforms of Heat-Stressed Broiler Chicken

**DOI:** 10.3390/vetsci10120678

**Published:** 2023-11-27

**Authors:** Youssef A. Attia, Reda A. Hassan, Nicola Francesco Addeo, Fulvia Bovera, Rashed A. Alhotan, Adel D. Al-qurashi, Hani H. Al-Baadani, Mohamed A. Al-Banoby, Asmaa F. Khafaga, Wolfgang Eisenreich, Awad A. Shehata, Shereen Basiouni

**Affiliations:** 1Sustainable Agriculture Production Research Group, Agriculture Department, Faculty of Environmental Sciences, King Abdulaziz University, Jeddah 21589, Saudi Arabia; aalqurashi@kau.edu.sa; 2Department of Animal and Poultry Production, Faculty of Agriculture, Damanhour University, Damanhour 22516, Egypt; 3Department of Poultry Nutrition, Animal Production Institute, Agricultural Research Center, Dokki, Giza 3751310, Egypt; redaalihasan@yahoo.com; 4Department of Veterinary Medicine and Animal Production, University of Napoli Federico II, Via F. Delpino 1, 80137 Napoli, Italy; nicolafrancesco.addeo@unina.it (N.F.A.); bovera@unina.it (F.B.); 5Department of Animal Production, College of Food and Agricultural Sciences, King Saud University, Riyadh 11451, Saudi Arabia; ralhotan@ksu.edu.sa (R.A.A.); hsaeed@ksu.edu.sa (H.H.A.-B.); 6Al-Shamel Animal Feed Factory, Industrial Area, Hail 55411, Saudi Arabia; 7Department of Pathology, Faculty of Veterinary Medicine, Alexandria University, Alexandria 21521, Egypt; 8Department of Chemistry, TUM School of Natural Sciences, Bavarian NMR Center (BNMRZ), Structural Membrane Biochemistry, Technical University of Munich, 85748 Garching, Germany; wolfgang.eisenreich@mytum.de; 9Clinical Pathology Department, Faculty of Veterinary Medicine, Benha University, Moshtohor, Toukh 13736, Egypt; 10Cilia Cell Biology, Institute of Molecular Physiology, Johannes-Gutenberg University, 55131 Mainz, Germany

**Keywords:** broiler, *Spirulina*, allicin, antioxidant, immunity, gut microbiota

## Abstract

**Simple Summary:**

Heat stress is a critical problem in poultry farming. In the current study, we evaluated the potential use of *Spirulina platensis* (SP) and *Allium sativum* (garlic powder, GP) to alleviate the adverse effects of heat stress in commercial broilers. Our findings suggest that supplementing the diet of heat-stressed broiler chickens with SP and/or GP can assist in mitigating the harmful impacts of heat stress on chickens’ performance. This includes improvements in productive traits, biochemical parameters, gut microbiota, and immunity. Furthermore, the simultaneous supplementation of SP and GP showed a synergistic effect, leading to even greater benefits.

**Abstract:**

This study aims to evaluate the effectiveness of the dietary addition of *Spirulina platensis* (SP) and/or garlic powder (GP) on heat-stressed broiler chickens. For this purpose, 600 Ross-308 broiler chicks were allocated at 22 days of age into five groups (G1–G5), each comprising six groups of 20 birds each. Chickens kept in G1 (negative control) were fed a basal diet and raised at 26 ± 1 °C. Chickens kept in G2 to G5 were exposed to periodic heat stress (35 ± 1 °C for 9 h/day) from 22 to 35 days old. Chickens in G2 (positive control) were provided a basal diet, while G3, G4, and G5 were fed a basal diet enriched with SP (1 g/kg diet), GP (200 mg/kg diet), or SP/GP (1 g SP/kg + 200 mg GP/kg diet), respectively. The assessment parameters included the chickens’ performance, malondialdehyde and total antioxidant capacity, blood biochemistry, intestinal morphology, and modulation of lactobacilli and total coliforms in the intestinal microbiota. Our findings demonstrated that supplementing heat-stressed chickens with SP and/or GP significantly mitigated the negative effects on the European production efficiency index (EPEF), survival rate, cholesterol profile, and oxidative stress markers. Chickens supplemented with GP and/or SP exhibited significantly better EPEF and survivability rates. Heat stress had a significant impact on both the gut structure and gut microbiota. However, SP and/or GP supplementation improved the gut morphology, significantly increased the intestinal lactobacilli, and reduced the coliform contents. It was also found that the simultaneous feeding of SP and GP led to even higher recovery levels with improved lipid metabolites, immunity, and oxidative status. Overall, supplementing chickens with SP and/or GP can alleviate the negative effects of heat stress.

## 1. Introduction

Chronic stress, known as the “secret killer” in poultry, results from commercial chicken farming, intestinal microbiota imbalance, mycotoxins, diet-mediated oxidative stress, environmental stressors, and nutritional imbalances [1,2,3]. Heat stress is a serious problem in poultry [1], primarily because of their feathers, elevated core body temperature, increased metabolic activity, and absence of sweat glands [2,4,5]. It increases the risk of infections and can negatively affect the body’s ability to function properly. This includes hindering the absorption of nutrients, digestion of food, water absorption, and other physiological processes [1,2,3]. Several health problems have also been documented as a consequence of reduced feed consumption and impairment of the immune system [6,7,8]. Thus, prolonged exposure to heat stress can weaken the immune system, making it more susceptible to various immunological diseases [9]. Heat stress can also induce oxidative stress, which leads to apoptosis and increases the intestinal barrier permeability [10]. In extreme cases, heat stress can result in hyperthermia and increased mortality rates [11].

Indeed, mitigating heat stress is important for animal welfare, performance, and economic sustainability in the livestock industry [12]. Several strategies have been proposed to mitigate the negative impacts of heat stress, including nutritional interventions such as feed restriction, wet or dual feeding, vitamins, minerals, osmolytes, and phytogenic substances [13]. In recent years, there has been a great interest in exploring sustainable natural alternative feed additives for poultry to enhance their productivity and health [1,2,3]. The diverse pharmacological and nutritional benefits of algae (*Spirulina* spp.) and garlic (*Allium sativum*) [14,15] have raised the question of their potential use in alleviating the adverse effects of heat stress [16].

*Spirulina platensis* (SP), blue-green algae rich in nutrients, can grow in freshwater and saltwater environments. It is a single-celled microalga and is a source of high nutrient content, making it a valuable feed supplement for poultry. It also contains various beneficial ingredients, including minerals, vitamins, phytopigments, and proteins [14,15]. Additionally, SP has high concentrations of bioactive substances, such as phycocyanins, steroids, saponins, chlorophyll, carotene, flavonoids, triterpenoids, and phenolic acids, which exhibit wide biological effects, including antibacterial, antiviral, anti-inflammatory, hepatoprotective, and antioxidant activities [17,18,19,20,21]. This renders it valuable to avian production, improving productivity and profitability [14,19,20]. Together, these many pathways could allow SP to effectively reduce the adverse effects of heat stress on chickens [20] and enhance animal performance and immune modulation [22].

Garlic (*Allium sativum*), a perennial blooming plant, contains a high oil content comprising sulfur compounds (diallyl sulfide and allicin), which have anti-inflammatory, antioxidant, anti-atherosclerotic, immune-modulating [23], and antibacterial properties [24]. Additionally, the potential benefits of garlic powder (GP) in poultry feeding have been declared to enhance intestinal health and boost the immune response [25], emphasizing its potential role in alleviating the adverse effects of heat stress.

Taken together, both SP and GP possess several beneficial properties, particularly anti-inflammatory and antioxidant effects, which make them promising candidates for alleviating chronic stress, such as heat stress. Accordingly, this study aims to investigate whether SP and/or GP can reduce the negative effects of heat stress in commercial broiler chickens.

## 2. Materials and Methods

### 2.1. Garlic Powder and Spirulina platensis

The GP was purchased from a local market (Gloryvet Company, Cairo, Egypt). The mixture containing garlic is sold under the trade name “Garllicin”, and each kilogram contains 750 g of a carrier (silicon dioxide) and 250 g of Allicin (C_6_H_10_S_3_) from GP. It is recommended to be used at a 50–200 g/t dose.

The dried powder of SP was provided by the Desert Research Center, Cairo, Egypt. Since the bioactive compounds in this product were unknown, the total flavonoid concentration and antioxidant properties were determined. Briefly, the Folin–Ciocalteu assay was used to quantify the dried SP powder’s total phenols, according to Kalagatur et al. [26]. The spectrophotometric method was conducted using a Helios Epsilon spectrophotometer (Thermo Spectronic, Rochester, NY, USA) according to Zhishen et al. [27]. The total phenols were expressed as mg gallic acid equivalents/100 g, while the total flavonoids were given as mg quercetin equivalents/100 g.

The antioxidant effect of SP dry powder was determined using the scavenger 1,1-diphenyl-2-picrylhydrazyl (DPPH) radical [28]. Briefly, after adding 2 mL of ethanolic DPPH solution to 1 g of SP, the mixture was incubated in the dark for 30 min. The absorbance was measured at λ max 517 nm, and the DPPH scavenging activity was calculated and expressed as a %.

The antioxidant capacity of SP was determined according to Prieto et al. [29]. Briefly, 10 mL of a solution containing 0.6 M sulfuric acid (H_2_SO_4_), 4 mM ammonium molybdate ((NH_4_)_2_MoO_4_), and 28 mM sodium phosphate (Na_3_PO_4_) were added to 1 mL of the sample, followed by thoroughly mixing and heating at 95 °C for 90 min. After cooling, the optical density was measured at OD_695_ nm. The results were expressed as ascorbic acid equivalents (mg/100 g).

### 2.2. Chickens and Experimental Design

This study followed all necessary methods and procedures approved by the Scientific Research Ethical Committee of King Saud University (KSU-SE-21-47). The experiment involved 600 healthy unsexed Ross-308 broiler chicks. The relative humidity (RH) was maintained at 65% and the temperature at 35 ± 1 °C on the first day of the trial, with a gradual decrease of 3 °C per week until it reached 26 °C on day 21. From day 22 to 35 of age, the chicks were divided randomly into five groups, with 20 birds in each replicate (six replicates per treatment). The chickens housed in Group 1 (G1) were kept at a temperature of 26 °C and a RH of 65 ± 5% as the negative control. However, the chickens in G2-G5 were subjected to heat stress at 35 ± 1 °C and 75 ± 5% RH, respectively, from 9:00 to 18:00 (9 h/day). The temperature and RH were 26 °C and 65 ± 5% for the remaining 15 h.

The birds were housed in floor pens (1 × 2 m^2^) ten birds/m^2^. The poultry house was partitioned into two sections: one for the thermoneutral treatment and the other for G2–G5. The walls that separated the sections were made of wood, and there was adequate distance between each portion. Gas heaters were used as the heat source, and a thermometer was employed to monitor the temperature inside the pens. The temperature was measured several times daily at different locations in the pens. The diet was formulated to meet chicks’ nutritional requirements [30]. The feeding system followed a two-phase approach, with starter grower feed provided from days 1 to 21 and finisher feed from days 22 to 35, as outlined in Table 1. Fresh water and mash feed were available to the birds in unlimited quantities. The chickens were vulnerable to light 24 h per day for the first week, then reduced to 22 h. Additionally, the chickens were vaccinated using inactivated Newcastle disease (NDV) and inactivated avian influenza (H9N1) vaccines subcutaneously on day 7, and they received the infectious bursal disease (IBD) vaccine on day 14. The experimental design for the assessment of SP and/or GP is illustrated in Figure 1.

### 2.3. Assessment Parameters

#### 2.3.1. Mortality and Growth Performance

The mortality was recorded daily, and the survival rate was calculated against the total number of chickens per treatment at the initiation of the trial. The body weight (BW), daily feed intake (DFI), and feed conversion ratio (FCR) were assessed. The DFI was calculated by subtracting the amount of feed (in grams) consumed on the last day from the amount consumed on the first day. The FCR was adjusted for mortality according to Nusairat et al. [31]. The European production efficiency index was also estimated: EPEI = survivability% × average body weight (kg)/market age (day) × FCR (kg feed/kg gain) × 100 [32].

#### 2.3.2. Hematogram and Biochemical Parameters

At 35 days old, blood samples (n = 6 samples/group) were gathered and divided into two parts. One part was collected in disodium ethylene diamine tetra-acetic acid tubes for the hematological analysis, including red blood cell counts (RBCs), white blood cells (WBCs), heterophils, lymphocytes, monocytes, and basophils. The second part was used for serum separation by centrifugation at 3000 rpm/min for 15 min; the serum samples were stored at -20 °C for biochemical examination. The serum total protein, albumin, triglycerides, cholesterol, low-density lipoprotein (LDL), high-density lipoprotein (HDL), uric acid, creatinine, alanine aminotransferase (ALT), and aspartate aminotransferase (AST) were evaluated spectrophotometrically (Stanbio Laboratory, Boerne, TX, USA). A radioimmunoassay kit (Institute of Isotopes Co., Ltd., Budapest, Hungary) was used to assess the serum triiodothyronine (T3) and thyroxine (T4) hormone concentrations. A chicken corticosterone ELISA kit (ZellBio, GmbH, Ulm, Germany) was used to measure the serum corticosterone concentration. The antioxidant markers, malondialdehyde (MDA), and total antioxidant capacity (TAC) were measured using commercial kits (Spinreact Co., Girona, Spain). Antibody titers for IBD and NDV were determined using ELISA (Indical Bioscience GmbH, Leipzig, Germany) and the hemagglutination inhibition test (HI), respectively. Chicken-specific IgA, IgM, and IgY were quantified using ELISA kits (Bethyl Laboratories Inc., Montgomery, TX, USA). To determine the heterophil-to-lymphocyte (H/L) ratio, fresh blood samples were smeared, fixed with methanol, and stained with Giemsa dye in a Coplin jar, followed by the counting of 60 cells per slide using a light microscope [33].

#### 2.3.3. Lymphoid Organs and Intestinal Morphology

At 35 days old, six chicks/group were slaughtered, and the thymus, spleen, and bursa of Fabricius were excised and weighed. The distal portions of the ileum (3 cm) were fixed in 10% formalin solution (five samples per treatment) after the digesta was flushed out with phosphate buffer saline for the histomorphological investigation. The ileal was cut into slides using a rotary microtome (Thermo Shandon, Cheshire, United Kingdom) and inspected under an optical microscope. The crypt depth (CD) and villus height (VH) were estimated [20].

#### 2.3.4. Modulatory Effect on Lactobacilli and Total Coliforms Microflora

Six samples per treatment were collected from the cecum at 35 days old. The bacterial count was performed using 0.1 g of the sample suspended in distilled water, homogenized, and diluted from 10^−1^ to 10^−10^. Then, 100 μL of the diluted solution was evenly spread on de Man–Rogosa–Sharpe agar (MRS agar, Oxoid, Germany) and MacConkey agar (Oxoid, Germany) for the counting of lactobacilli and total coliforms, respectively. The MRS plates were incubated at 37 °C for 48 h in anaerobic conditions, while the MacConkey plates were incubated for 24 h at 37 °C aerobically [34]. The colony-forming units (CFU) were expressed as log_10_ per g of feces or cecal content.

### 2.4. Statistical Analysis

Statistical analysis software (SAS, 2004) was used according to one-way ANOVA, using the replicate as the experimental unit. The *p*-value < 0.05 was considered for significance. Further, Tukey’s post hoc test was used for means comparisons. The normality of the data was tested before running the statistical analyses using the Shapiro-Wilk test of normality [35].

## 3. Results

### 3.1. Analysis of S. platensis

Table 2 illustrates the bioactive compounds found in SP, which include significant amounts of total phenols and flavonoids. The results revealed that SP has a high DPPH scavenging activity of 90% and a total antioxidant capacity of 46.50 mg ascorbic acid equivalents/g.

### 3.2. Performance Parameters

Table 3 shows the effects of the dietary SP and/or GP on the EPEI and survival rates of chickens subjected to periodic heat stress. Heat stress exposure had a negative influence on the EPEI. Chickens kept in G1 had a significantly higher EPEI than G2 (heat-stressed non-treated). SP and/or GP supplementations markedly increased the EPEI compared to G2. The most effective EPEI was obtained in chickens treated with SP and GP (G5). Table 3 indicates that the heat-stressed chicks exhibited higher survival rates, with a 20% decrease compared to G1. Moreover, the survival rate of chicks fed SP and/or GP was increased compared to the chickens kept in G2. The survival % was 100%, 80%, 98.3%, 98.3%, and 100% in G1, G2, G3, G4, and G5, respectively.

### 3.3. Lymphoid Organ Weights and Hematological Blood Variables

Heat stress impacted the lymphoid organ weights and hematological parameters. There was a significant (*p* < 0.05) decreased weight of the bursa of Fabricius, spleen, and thymus. The red blood cells, white blood cell count, and lymphocyte % were also significantly decreased. There was also a higher number of heterophils (H%), H/L ratio, and cortisone than the thermoneutral control broiler chicks (Table 4). Supplementing the diet with SP and/or GP completely alleviated the adverse effects of heat stress on the spleen and thymus weights and partially on the bursa. There was a partial recovery in most of the hematological parameters due to SP and/or GP supplementations, but the addition of SP with GP induced a complete recovery in lymphocytes. There were notable variations in the corticosterone levels among the various regimens (Table 4). Chickens reared in heat stress conditions (G2) had significantly greater corticosterone concentrations than G1. However, broilers given SP or GP, particularly SP/GP (G5) of the feed, exhibited significantly reduced corticosterone concentrations during heat stress compared to G2.

### 3.4. Serum Metabolites and Oxidative Blood Markers

The effects of SP and GP on the blood parameters are shown in Table 5. The results showed that blood metabolites were significantly affected by exposure to heat stress. The lipid analysis revealed a significant elevation in triglycerides, cholesterol, and LDL levels but decreased HDL levels in chickens that experienced heat stress and were not treated (G2) compared to the SP- and/or GP-treated chickens. The serum levels of AST, ALT, creatinine, and uric acid were also significantly higher in G2 compared to G1, but the uric acid-to-creatinine ratio decreased. The negative impact of heat stress on lipid metabolites, liver function, and kidney function was ameliorated by SP and/or GP supplementation. SP and/or GP supplementation showed a complete recovery in cholesterol, LDL, triglycerides, creatinine, uric acid concentrations, and AST activity while increasing the HDL levels compared to the G1 and G2 groups.

The broiler chicks’ oxidative status was negatively impacted by exposure to heat stress, showing an elevation in their MDA levels and low serum TAC concentrations (Table 5). SP and/or GP supplementation reduced the concentrations of MDA and increased the TAC and TAC/MDA ratio, highlighting the amelioration of the detrimental impacts of heat stress on their oxidative state.

### 3.5. Intestinal Microbiota and Morphology

The total counts of lactobacilli and coliforms are presented in Table 6. The high levels of coliforms showed a considerable rise due to heat stress exposure (G2). However, the *Lactobacillus* spp. count was significantly lower compared to G2. Additionally, supplementation with SP and/or GP decreased the harmful effects of heat stress on bacterial enumeration by reducing the coliforms and boosting the enumeration of beneficial *Lactobacillus* spp.

Heat stress negatively impacted the intestinal morphology of the birds (G2), in which the VH and VH:CD ratio decreased while the CD increased (Table 6 and Figure 2). However, heat-stressed birds fed SP and/or GP (G3–G5) had intestinal morphology similar to the chickens raised under a thermoneutral environment (G1).

### 3.6. Immune Response

The immunological parameters are shown in Table 7. It was found that IgM, IgA, and IgY were significantly reduced in heat-stressed chickens (G2) compared to the other groups. However, SP and/or GP supplementation mitigated the negative effects of heat stress on Igs (IgM, IgA, and IgY), as well as antibodies against NDV and IBD.

## 4. Discussion

Heat stress can result in oxidative stress, inflammation, and alterations in the gut microbiota composition in birds. These changes can render the bird more susceptible to pathogens and hurt its performance [1,2]. Natural feed additives are of the utmost importance in poultry production due to their beneficial biological properties, including antioxidants, anti-inflammatory, immunostimulatory, and antimicrobial characteristics, all of which can assist in addressing various health-related concerns [2,4]. Therefore, in the present study, we assessed the effectiveness of SP and/or GP in reducing the damaging effects of heat stress.

The SP analysis found that the sample contained a significant number of total phenols (15.85 mg/g) and total flavonoids (1.75 mg/g). The DPPH scavenging activity was 90%, and the total antioxidant capacity was 46.5 mg/g. Garlic contains superoxide dismutase, enzymes, glycosides, and allicin and organic selenium [36]. This makes them potential candidates to be used as alternative strategies against heat stress.

Heat stress has been proven to reduce the EPEI, a reliable performance index based on the daily gain, survival rate (20% mortality), and FCR. However, SP and/or GP can significantly improve the EPEI. Our findings are consistent with recent research indicating that supplementing broilers with SP can enhance their growth performance [37]. It is believed that this positive effect can be attributed to the high-quality protein and growth-promoting substances present in SP, as well as the bioactive substances in both SP and GP [38,39,40].

Nonetheless, heat stress has been linked to increased heterophils and decreased lymphocytes, resulting in an elevated H/L ratio, which is an indicator of heat stress [41]. Interestingly, the hemoglobin, RBCs, WBCs, lymphocyte, heterophil, and H/L ratio were significantly improved by SP and/or GP supplementation. The improvement in RBCs with GP supplementation may be due to the possibility that GP directly increases kidney erythropoietin production and secretion, which results in erythrocyte synthesis [42]. According to our study, SP also enhanced the birds’ RBC and WBC counts. This may also be attributed to the SP polysaccharides that can increase the number of nucleated cells in the bone marrow of dogs and the number of RBCs, WBCs, and blood Hgb [43]. An additional interesting biological property of SP includes its ability to prevent anemia due to its abundance of iron and vitamin contents [44]. GP significantly reduced the H/L ratio; this outcome was consistent with the observations made by [45,46]. Dorhoi et al. found that GP supplementation tends to increase the uptake of spleen RBCs [47].

Within the scope of this study, exposure to heat stress increased the serum cholesterol, LDL, and triglyceride levels, while the HDL and Hgb levels decreased. These outcomes can be partially explained by the stimulation of the hypothalamic–pituitary–adrenal axis and the release of glucocorticoids [48]. Additionally, heat stress increases the serum triglyceride and cholesterol levels; liver triglyceride synthesis; and metabolites involving glucose, amino acids, and lipids [49]. Supplementing chickens with SP lowered their cholesterol, LDL, total lipids, and triglyceride levels. This may be due to the antioxidant properties of SP, such as phycocyanin, phenolic compounds, and polyunsaturated fatty acids [50]. GP may also have a positive effect on lipid metabolites by inhibiting the activity of lipogenic and cholesterol-genic enzymes, such as malic enzyme, fatty acid synthases, and glucose-6-phosphatase dehydrogenase, resulting in hypocholesterolemia and hypolipidemic action [51].

Moreover, allicin contributes to a decrease in cholesterol [52]. The most effective enzyme in the production of cholesterol and lipids, hydroxymethlygutaryl-CoA reductase, is inhibited by allicin [53]. The reduction in hepatic enzymes by GP supplementation might result from the antioxidant effects and stabilization of the liver cell membrane and inhibition of the negative effects of free radicals and hazardous agents [54]. Thus, free radicals could cause serious damage to liver cells, releasing three enzymes, AST, ALT, and alkaline phosphatase, into the bloodstream.

Commercial broiler chickens are particularly susceptible to oxidative stress, because they have been genetically bred for rapid growth. High temperatures in broiler farming induce oxidative stress, causing negative biological effects, illness, and poor growth performance [55]. In the present study, SP and/or GP resulted in a considerable rise in the TAC and a decrease in MDA. Reduced MDA levels showed that the cell was not under oxidative stress conditions. The following mechanisms can explain the antioxidant effect of SP and GP: (i) SP is a great source of natural antioxidants such as selenium, carotene, tocopherol, polypeptide pigment, and phenolic compounds [56], and (ii) GP contains allicin, which has potent antiradical mechanisms and reduces iron [57,58]. The combination with SP could enhance heat-stressed birds’ antioxidant status.

Heat stress has been shown to impact hens’ intestinal morphology and barrier integrity, which reduces their capability to digest and absorb food and increases their permeability to luminal antigens and toxins [59,60]. Broilers administered SP and/or GP showed a considerable improvement in intestinal morphology. Additionally, a synergistic effect was obtained when SP/GP was observed. The enhanced colonization of beneficial bacteria may be the reason for the favorable impact of SP and/or GP on the gut morphology [61]. Also, the small intestine’s length and width may increase due to garlic’s antimicrobial activity. Meanwhile, according to Rehman and Munir [42], phenolic derivatives can influence the intestinal microbial ecology in chickens and lead to improvements in the intestinal flora of broilers.

In this study, heat stress led to changes in the composition of the intestinal microbiota by increasing the number of coliforms and decreasing the abundance of nonpathogenic bacteria such as *Lactobacillus*. These findings are consistent with previous research [60]. However, SP and/or GP could alleviate this negative effect. Our findings are consistent with previous research indicating that SP increases *Lactobacillus* spp. and decreases *E. coli* populations [62,63]. The intestinal microbiota can be positively affected by SP and GP due to their antibacterial properties. SP contains polysaccharides, which have a prebiotic effect [64]. Additionally, SP contains active ingredients such as tocopherols and C-phycocyanin that have antimicrobial effects against various bacteria, including *Pseudomonas* spp., *E.coli*, *Klebsiella pneumonia*, *Enterobacter* spp., *Proteus vulgaris*, and *Salmonella* Typhi [65]. Garlic minimizes the number of harmful bacteria while increasing the number of beneficial bacteria [66]. Further assessments of the modulatory effects of SP and GP on other intestinal microbiota are still needed. However, the modulatory effect of GP was only studied on coliforms and *Lactobacillus* spp., which is a limitation of our study.

Indeed, the weights of the thymus, bursa, and spleen were significantly reduced due to heat stress. This could be because of a decreased feed intake, resulting in fewer nutrients for these organs to develop properly. However, when supplemented with SP and/or GP, the weights of the lymphoid organs (thymus, bursa, and spleen) increased. These findings are in line with previous studies [67,68]. Additionally, heat-stressed birds not fed GP or SP tended to have lower amounts of IgA, IgY, and IgM antibodies and lower titers against NDV and IBDV. This may be attributed to their lymphoid organs having lower relative weights, which could be linked to poor growth or caused by dysfunctions induced by heat stress, such as oxidative stress or impaired immune responses [8]. These results agree with those indicating that heat stress could reduce the IgA, IgM, and IgY antibody levels [69]. However, both SP and GP exhibited immunostimulatory effects that can be explained by (i) high Zn concentrations in SP could contribute to developing birds’ cellular immunity [45]; (ii) a modulatory effect on the intestinal lactobacilli and improvement of intestinal morphology. (iii) garlic stimulates lymphocyte and macrophage division and differentiation and activates phagocytosis, IL-2 synthesis, and interferon production [70]; and (iv) a potential synergistic effect between SP and GP supplementations can be shown, which may be explained by the antioxidant properties of SP or GP, which help to stabilize membranes, although its effect on corticosteroid release cannot be excluded. Increased corticosterone levels brought on by heat stress cause lymphocytes to vanish, which causes lymphopenia [71].

## 5. Conclusions

The intestinal villus height, Lactobacillus count, lymphoid organs, and immunity status of commercial broilers exposed to heat stress were improved by *Spirulina platensis* and/or garlic powder. Our suggestions adding 1g of *Spirulina platensis*/kg or 200 mg of garlic powder/kg diet. Furthermore, our findings indicated that *Spirulina platensis* and garlic powder exhibit synergistic effects in mitigating the harmful effects of heat stress. Further analysis of the bioactive substances of both SP and GP extracts using high-end analytics such as nuclear magnetic resonance and studying their mechanisms of action as antibacterial agents are still required.

## Figures and Tables

**Figure 1 vetsci-10-00678-f001:**
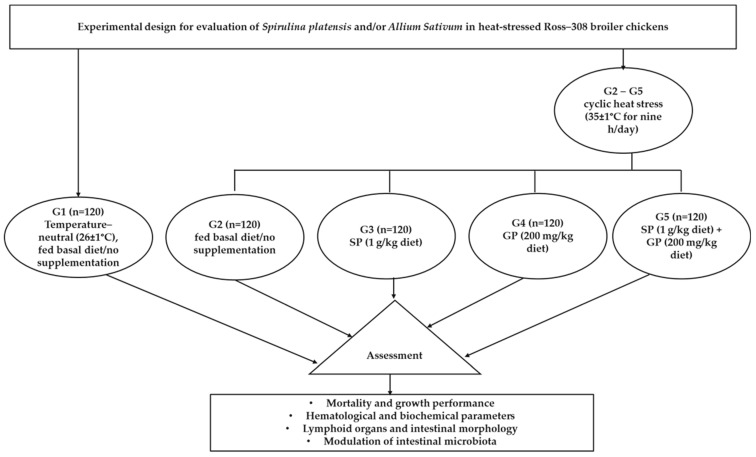
Experimental design for evaluating *S. platensis* (SP) and garlic powder (GP) in commercial broiler chickens subjected to heat stress.

**Figure 2 vetsci-10-00678-f002:**
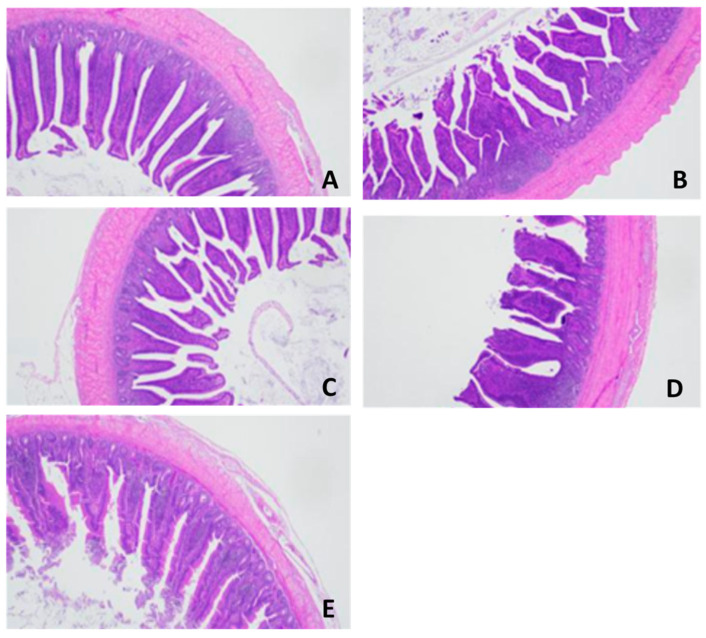
Representative photomicrographs for intestinal tissues (ilium) of chickens subjected to heat stress (**B**) and supplemented with GP (**C**), SP (**D**), or their combination (**E**) compared to the control chickens (H&E, ×10). (**A**) Control tissues revealed the normal histomorphology structure of the intestinal villi, associated crypt, tunica muscularis, and properly arranged columnar epithelium and goblet cells. (**B**) Tissues from chickens raised under heat stress showing necrosis of the villus epithelium and desquamation within the lumen. (**C**,**D**) Intestinal tissues from heat-stressed chickens treated with GP or SP showed mild improvement in the intestinal histological structure. (**E**) Tissues from heat-stressed chickens treated with a combination of GP and SP showed moderate improvement in the histologic structure of the intestinal villi and arrangement of the columnar epithelium and goblet cells.

**Table 1 vetsci-10-00678-t001:** Composition and calculated analysis of the used diets.

Ingredients (%)	Starter Grower (1–21 d)	Finisher (22–35 d)
Yellow corn	54.40	62.00
Soybean meal 44%	27.00	24.05
Corn gluten meal 60%	10.00	6.19
Soybean oil	4.55	4.00
Limestone	1.10	1.00
Dicalcium phosphate	2.20	2.05
Vitamins and minerals premix ^1^	0.30	0.30
DL-methionine	0.05	0.01
L-lysine (HCl)	0.15	0.15
NaCl	0.25	0.25
Total	100	100
Calculated analysis ^2^		
Crude protein (%)	23.03	20.02
Metabolizable energy (kcal/kg)	3204	3201
Calcium (%)	1.05	0.97
Available phosphorus (%)	0.45	0.42
Lysine (%)	1.14	1.03
Methionine (%)	0.52	0.41
Total sulfur amino acids (%)	0.90	0.73

^1^ Each 3 kg contains Vit A 12,000,000 IU, Vit D3 2,000,000 IU, Vit E 10 g, Vit K3 2 g, Vit B1 1 g, Vit B2 5 g, Vit B6 1.5 g, Vit B12 10 mg, nicotinic acid 30 g, pantothenic acid 10 g, folic acid 1 g, biotin 50 mg, choline chloride 250 g, iron 30 g, copper 10 g, zinc 50 g, manganese 60 g, iodine 1 g, selenium 0.1 g, cobalt 0.1 g, and calcium carbonate (CaCo_3_) as a carrier to 3 kg. NaCl = sodium chloride. ^2^ According to tables of NRC (1994) [30].

**Table 2 vetsci-10-00678-t002:** Bioactive components and antioxidant activities of *S. platensis*.

Component	Content
Total phenols, mg gallic acid equivalent/100 g	1585
Total flavonoids, mg quercetin equivalent/100 g	175
DPPH scavenging activity (%) *	90
Total antioxidant capacity (mg ascorbic acid equivalent/100 g)	46.50

* 1,1-diphenyl-2-picrylhydrazyl (DPPH) radical.

**Table 3 vetsci-10-00678-t003:** Effects of *S. platensis* (1 g/kg diet) and/or garlic powder (200 mg/kg diet) on broiler chickens’ performance subjected to heat stress from 22 to 35 days of age.

Parameters Traits ParaP	G1 (Negative Control)	Groups Raised under Heat Stress				SEM	*p*-Value
		G2	G3	G4	G5		
Survival %	100.0 ^a^	80.0 ^c^	98.3 ^b^	98.3 ^b^	100.0 ^a^	0.008	<0.001
Final body weight (BW) *	2135 ^a^	1850 ^c^	2035 ^b^	2020 ^b^	2100 ^a^	25.22	0.005
Daily feed intake (DFI) *	108	103.4	104.3	104.1	105.7	10.22	0.058
Feed conversion ratio (FCR) *	1.80 ^b^	2.00 ^a^	1.83 ^b^	1.84 ^b^	1.80 ^b^	0.257	0.028
EPEI **	338.1 ^a^	229.9 ^c^	311.5 ^b^	310.4 ^b^	334.1 ^a^	4.528	0.005

Means with different letters indicate significant differences (*p* < 0.05); G1 = fed the basal diet and kept in a temperature-neutral environment (negative control); G2 = fed the basal diet and kept under a heat stress environment (positive control); G3, G4, and G5 = fed the basal diet containing S. *platensis*, garlic powder, or both, respectively, and raised under a heat stress environment. SEM, standard error of the mean. * BW and FCR were assessed at 35 days old [16]; ** EPEI = European production efficiency index [32].

**Table 4 vetsci-10-00678-t004:** Effects of dietary supplementation with *S. platensis* (1 g/kg diet) and/or garlic powder (200 mg/kg diet) on broiler chickens’ lymphoid organs, hematological parameters, and stress indices subjected to heat stress at 22–35 days of age.

Parameters		G1	Groups Raised under Heat Stress				SEM	*p*-Value
			G2	G3	G4	G5		
Lymphoid organs	Bursa (%)	0.196 ^a^	0.135 ^c^	0.168 ^b^	0.177 ^b^	0.169 ^b^	0.025	0.024
	Spleen (%)	0.221 ^a^	0.155 ^c^	0.176 ^b^	0.185 ^b^	0.201 ^ab^	0.036	0.001
	Thymus (%)	0.425 ^a^	0.302 ^b^	0.411 ^a^	0.405 ^a^	0.415 ^a^	0.044	0.035
Hematological parameters	RBCs (×10^6^)	3.65 ^a^	2.70 ^d^	3.10b ^c^	3.05 ^c^	3.40 ^b^	0162	0.001
	WBCs (×10^3^)	4.12 ^a^	3.52 ^c^	4.00 ^b^	4.05 ^b^	4.10 ^b^	0.082	0.002
	Heterophils (%)	23.5 ^d^	31.6 ^a^	27.0 ^b^	25.6 ^c^	25.0 ^c^	0.530	0.036
	Lymphocytes (%)	76.3 ^a^	67.9 ^d^	72.7 ^c^	74.0 ^b^	74.6 ^ab^	0.426	0.001
	Monocyte (%)	0.16	0.35	0.23	0.30	0.25	0.122	0.625
	Basophil (%)	0.04	0.19	0.12	0.10	0.15	0.04	0.051
Stress indices	H/L ratio	0.308 ^c^	0.465 ^a^	0.372 ^b^	0.346 ^b^	0.335 ^bc^	0.02	0.025
	Corticosterone (nmol/L)	26.2 ^b^	29.2 ^a^	27.0 ^ab^	26.8 ^b^	26.2 ^b^	0.244	0.020

Means with different letters indicate significant differences (*p* < 0.05); G1 = fed the basal diet and raised in a temperature-neutral environment; G2 = fed the basal diet and raised under a heat stress environment; G3, G4, and G5 = fed the basal diet containing S. *platensis*, garlic powder, or both, respectively, and raised under a heat stress environment. SEM, standard error of the mean. RBCs, red blood cells; WBCs, white blood cells.

**Table 5 vetsci-10-00678-t005:** Effects of *S. platensis* (1 g/kg diet) and/or garlic powder (200 mg/kg diet) on some blood constituents, liver and renal functions, and antioxidants status of commercial broilers subjected to heat stress at 22–35 days of age.

Parameters		G1 (Negative Control)	Groups Raised under Heat Stress				SEM	*p*-Value
			G2	G3	G4	G5		
Lipid metabolites	Triglycerides (mg/dL)	179.1 ^b^	196.8 ^a^	180.2 ^b^	178.6 ^b^	177.8 ^b^	3.274	0.001
	Cholesterol (mg/dL)	190.2 ^bc^	244.5 ^a^	200.6 ^b^	195.8 ^bc^	188.5 ^c^	3.552	0.018
	LDL (mg/dL)	90.8 ^bc^	140.5 ^a^	98.7 ^b^	85.5 ^c^	85.0 ^c^	2.458	0.005
	HDL (mg/dL)	66.5 ^b^	60.8 ^c^	69.6 ^a^	67.5 ^b^	70.4 ^a^	2.118	0.004
	LDL/HDL	1.365 ^b^	2.310 ^a^	1.418 ^ab^	1.266 ^c^	1.207 ^c^	0.258	0001
Liver leakage enzymes	AST (U/L)	62.0 ^b^	66.8 ^a^	61.8 ^b^	61.5 ^b^	61.0 ^b^	1.008	0.025
	ALT (U/L)	70.2 ^b^	76.8 ^a^	69.8 ^b^	70.2 ^b^	69.0 ^b^	2.285	0.008
	ALT/AST ratio	1.132	1.149	1.129	1.173	1.131	0.055	0.325
Renal function	Creatinine (mg/dL)	0.288 ^b^	0.528 ^a^	0.278 ^b^	0.269 ^b^	0.275 ^b^	0.025	0.004
	Uric acid (mg/dL)	5.33 ^b^	6.75 ^a^	4.58 ^c^	5.00b ^c^	4.98b ^c^	0.145	0.025
	Uric acid/creatinine ratio	18.51 ^a^	12.78 ^c^	16.47 ^b^	18.59 ^a^	18.11 ^a^	1.025	0.004
Antioxidants status	MDA (nmol/mL)	1.96 ^c^	3.22 ^a^	2.55 ^b^	2.38 ^b^	2.21 ^b^	0.058	0.035
	TAC (U/mL)	8.28 ^a^	5.36 ^c^	7.65 ^b^	7.85 ^b^	8.10 ^a^	1.145	0.004
	TAC/MDA ratio	4.224 ^a^	1.664 ^c^	3.00 ^b^	3.298 ^ab^	3.665 ^ab^	0.854	0.002

Means with different letters indicate significant differences (*p* < 0.05); G1 = fed the basal diet and raised in a temperature-neutral environment; G2 = fed the basal diet and raised under a heat stress environment; G3, G4, and G5 = fed the basal diet supplemented with *S. platensis*, garlic powder, or both, respectively, and raised under a heat stress environment. LDL, low-density lipoprotein cholesterol; HDL, high-density lipoprotein cholesterol; AST, aspartate aminotransferase; ALT, alanine aminotransferase; T3, triiodothyronine; MDA, malondialdehyde; TAC, total antioxidant capacity.

**Table 6 vetsci-10-00678-t006:** Effects of *S. platensis* (1 g/kg diet) and/or garlic powder (200 mg/kg diet) on the bacterial count and intestinal morphology of commercial broilers subjected to heat stress at 22–35 days of age.

Parameters		G1 (Negative Control)	Groups Raised under Heat Stress				SEM	*p*-Value
			G2	G3	G4	G5		
*Cecal* bacterial count (log_10_ CFU/g)	Coliforms	5.20 ^b^	7.92 ^a^	5.05 ^b^	5.62 ^b^	4.80 ^c^	0.155	0.024
	*Lactobacillus*	6.05 ^b^	5.33 ^c^	7.51 ^a^	7.65 ^a^	7.00 ^a^	0.224	0.035
*Fecal* bacterial count (log_10_ CFU/g)	Coliforms	5.33 ^b^	7.80 ^a^	5.25 ^b^	5.30 ^b^	4.77 ^c^	0.08	0.005
	*Lactobacillus* spp.	6.25 ^b^	5.00 ^c^	6.52 ^a^	7.42 ^a^	7.85 ^a^	0.15	0.002
Intestinal morphology (µm)	Villus height (VH)	1345 ^a^	1288 ^c^	1320 ^b^	1308 ^b^	1330 ^b^	28.255	0.001
	Crypt depth (CD)	332 ^b^	355 ^a^	330 ^b^	339 ^b^	328 ^b^	2.552	0.003
	VH/CD	4.05 ^a^	3.63 ^b^	4.00 ^a^	3.86 ^a^	4.05 ^a^	0.445	0.025

Means with different letters indicate significant differences (*p* < 0.05); G1 = fed the basal diet and raised in a temperature-neutral environment; G2 = fed the basal diet and raised under a heat stress environment; G3, G4, and G5 = fed the basal diet supplemented with *S. platensis*, garlic powder, or both, respectively, and kept under heat stress; CFU, colony-forming unit.

**Table 7 vetsci-10-00678-t007:** Effects of *S. platensis* (1 g/kg diet) and/or garlic powder (200 mg/kg diet) on the immunoglobulin and immune response of commercial broilers kept under heat stress at 22–35 days of age.

Parameters		Control G1	Groups Raised under Heat Stress				SEM	*p*-Value
			G2	G3	G4	G5		
Immunoglobulin (Ig)	IgM (mg/dL)	195 ^a^	96 ^c^	145 ^b^	179 ^ab^	180 ^ab^	4.025	0.004
	IgY (mg/dL)	819 ^a^	544 ^c^	725 ^b^	760 ^ab^	781 ^ab^	5.224	0.026
	IgA (mg/dL)	250	248	257	238	250	5.125	0.485
Immune response	NDV (Log_2_)	8.25 ^a^	6.45 ^c^	7.35 ^b^	7.44 ^b^	8.05 ^ab^	0.458	0.025
	IBD (Log_2_)	6.55 ^a^	4.28 ^c^	5.98 ^b^	6.08 ^ab^	6.38 ^a^	0.342	0.001

Means with different letters indicate significant differences (*p* < 0.05); G1 = fed the basal diet and raised in a temperature-neutral environment; G2 = fed the basal diet and raised under a heat stress environment; G3, G4, and G5 = fed the basal diet supplemented with *Spirulina platensis*, garlic powder, or both, respectively, and raised under a heat stress environment; NDV = Newcastle disease virus; IBD = infectious bursal disease.

## Data Availability

The data presented in this study are available on request from the corresponding author.

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
