# Peer review of "Effects of *Spirulina platensis* and/or *Allium sativum* on Antioxidant Status, Immune Response, Gut Morphology, and Intestinal Lactobacilli and Coliforms of Heat-Stressed Broiler Chicken"

_vetsci, 2023, doi:10.3390/vetsci10120678_

Round 1

Reviewer 1 Report

Comments and Suggestions for Authors

Dear authors,

Your paper: “Effects of Spirulina platensis and/or Allium Sativum on Antioxidant Status, Immune Response, Gut Morphology, and Intestinal Microbial of Heat-Stressed Broiler Chicken”, investigates the potential use of Spirulina platensis and Allium sativum to mitigate the negative impacts of heat stress on broiler chickens' performance, immune response, oxidative status, and intestinal microbiota. I found it very interesting because it highlights the benefits of using natural solutions to alleviate heat stress without negative effects.

I recommend the paper to be published after minor corrections.

Line spacing on Introduction chapter seems different from other chapters.

Raw 31: Heat stress can potentially mitigate the adverse effects on chicken performance. Regrettably, no specific performance parameters are mentioned in the paper. It would be helpful to include these parameters for context and analysis.

Raw 65: Please refrain from redundant phrasing. Rephrase to avoid repetition of “and”

Raw 71: If you began by including Latin names in brackets, kindly do the same for Spirulina spp. For consistency, you can use "Spirulina (Spirulina spp.)" or "Spirulina platensis." Please maintain this uniform approach.

 Raw 174: Please mention how many RPM, for how many minutes were the serum /plasma samples centrifugated?

Raw 222: It appears insufficient to exclusively discuss productive performances without including essential production parameters like initial/final body weight (BW), daily feed intake (DFI), and feed conversion ratio (FCR), separately within the table. These production parameters should be added for a comprehensive analysis.

Raw 233, Table 3: Please provide an explanation of the acronym "EPEI" as a footnote in Table 3 for clarity.

Raw 278: Add a space between the title of the tables and the preceding text.

Figure 1: Ensure uniformity in font and color for capital letters in Figure 1, the 5th photo, particularly in the right corner.

Raw 501: The font used for the conclusions section appears different from the rest of the text. Please make sure the font is consistent throughout the document.

Please kindly to add the limitation of this experiment.

Line spacing on Introduction chapter seems different from other chapters.

Raw 31: Heat stress can potentially mitigate the adverse effects on chicken performance. Regrettably, no specific performance parameters are mentioned in the paper. It would be helpful to include these parameters for context and analysis.

Raw 65: Please refrain from redundant phrasing. Rephrase to avoid repetition of “and”

Raw 71: If you began by including Latin names in brackets, kindly do the same for Spirulina spp. For consistency, you can use "Spirulina (Spirulina spp.)" or "Spirulina platensis." Please maintain this uniform approach.

 Raw 174: Please mention how many RPM, for how many minutes were the serum /plasma samples centrifugated?

Raw 222: It appears insufficient to exclusively discuss productive performances without including essential production parameters like initial/final body weight (BW), daily feed intake (DFI), and feed conversion ratio (FCR), separately within the table. These production parameters should be added for a comprehensive analysis.

Raw 233, Table 3: Please provide an explanation of the acronym "EPEI" as a footnote in Table 3 for clarity.

Raw 278: Add a space between the title of the tables and the preceding text.

Figure 1: Ensure uniformity in font and color for capital letters in Figure 1, the 5th photo, particularly in the right corner.

Raw 501: The font used for the conclusions section appears different from the rest of the text. Please make sure the font is consistent throughout the document.

Please kindly to add the limitation of this experiment.

Author Response

Your paper: “Effects of Spirulina platensis and/or Allium Sativum on Antioxidant Status, Immune Response, Gut Morphology, and Intestinal Microbial of Heat-Stressed Broiler Chicken”, investigates the potential use of Spirulina platensis and Allium sativum to mitigate the negative impacts of heat stress on broiler chickens' performance, immune response, oxidative status, and intestinal microbiota. I found it very interesting because it highlights the benefits of using natural solutions to alleviate heat stress without negative effects.

I recommend the paper to be published after minor corrections.

Response:  We are grateful to the reviewers for reviewing our manuscript and providing insightful comments. We have diligently addressed all the feedback received in this version, as can be seen in both this rebuttal and the revised manuscript. Thanks a lot for these comments that improved our manuscript.

Line spacing on Introduction chapter seems different from other chapters.

Response: Thank you very much for this comment; we revised it as kindly recommended. Please refer to the introduction part, Lines 65-110.

Raw 31: Heat stress can potentially mitigate the adverse effects on chicken performance. Regrettably, no specific performance parameters are mentioned in the paper. It would be helpful to include these parameters for context and analysis.

Response: Thank you very much. The productive traits [body weight (BW), daily feed intake (DFI), and feed conversion ratio (FCR)], See table 3.  

Raw 65: Please refrain from redundant phrasing. Rephrase to avoid repetition of “and”

Response: Thanks a lot. This sentence was paraphrased to be: Heat stress can also induce oxidative stress, which leads to apoptosis and increases the intestinal barrier permeability [10].  Lines 75-76.

Raw 71: If you began by including Latin names in brackets, kindly do the same for Spirulina spp. For consistency, you can use "Spirulina (Spirulina spp.)" or "Spirulina platensis." Please maintain this uniform approach.

Response: Thanks a lot. This sentence was revised as kindly recommended to be: The diverse pharmacological and nutritional benefits of algae (Spirulina spp.) and garlic (Allium sativum) [14,15] raised the question of their potential use in alleviating the adverse effects of heat stress [16]. Lines 83-86.

 Raw 174: Please mention how many RPM, for how many minutes were the serum /plasma samples centrifugated?

Response: Thanks a lot. To obtain the sera, blood samples were centrifuged at 3000 rpm/min for 15 min. Line 187.

Raw 222: It appears insufficient to exclusively discuss productive performances without including essential production parameters like initial/final body weight (BW), daily feed intake (DFI), and feed conversion ratio (FCR), separately within the table. These production parameters should be added for a comprehensive analysis.

Response: Highly appreciated. These parameters were added to the table as kindly recommended and discussed. Table 3.

Raw 233, Table 3: Please provide an explanation of the acronym "EPEI" as a footnote in Table 3 for clarity.

Response: Thanks a lot.  EPEI = * European production efficiency index. Line 250

Raw 278: Add a space between the title of the tables and the preceding text.

Response: Done as kindly recommended, Thank you very much

Figure 1: Ensure uniformity in font and color for capital letters in Figure 1, the 5th photo, particularly in the right corner.

Response: Both Figures 1 and 2 were revised, Thank you very much

Raw 501: The font used for the conclusions section appears different from the rest of the text. Please make sure the font is consistent throughout the document.

Response: Revised as kindly recommended, Thank you very much

Please kindly -to add the limitation of this experiment.

Response: Thank you very much. The limitations of the study were added in the discussion.  However, the modulatory effect of GP was only studied on coliforms and Lactobacillus spp., which is a limitation of our study.  Lines 447-449.

Reviewer 2 Report

Comments and Suggestions for Authors

The general perception is this manuscript was hurriedly written and has poor
formatting and descriptions. The design of the study is good and important for science. However, the authors must have to work on instruction, mainly on why mitigation of heat stress is important, what was done in the past, and what is new in this study. Also please check the typos and reference style.

Specific comments:

1) The growth performance was not included in the manuscript

2) What was the percentage of mortality? How body weight and FCR was adjusted after mortality?

3) Define which part of the ileum was collected

4) Specify whether or not the digesta was flushed out from the intestinal sections before fixing in formalin

5) Be consistent with either P<0.05 or P≤0.05

6) "Tukey post-hoc test" not Tukey post-hock test

7) Decisions on improved gut microbiota were made based on only Lactobacillus and Coliform bacteria, so I suggest changing the more specific title.

Comments on the Quality of English Language

Need major improvements.

Author Response

Reviewer 2

The general perception is this manuscript was hurriedly written and has poor
formatting and descriptions. The design of the study is good and important for science. However, the authors must have to work on instruction, mainly on why mitigation of heat stress is important, what was done in the past, and what is new in this study. Also please check the typos and reference style.

Response: Thank you very much for your comments. They have significantly enhanced the quality of our manuscript. The introduction was improved, and new paragraphs were added.

Specific comments:

  • The growth performance was not included in the manuscript

Response: Thanks a lot. Done as kindly recommended, Please see Table 3 and the discussion

  • What was the percentage of mortality? How body weight and FCR was adjusted after mortality?

Response: Thanks a lot. The survival % is shown in Table 3. The FCR was adjusted for mortality according to Nusairat et al. [31]. Line 179-180.

3) Define which part of the ileum was collected

Response: Thank you very much, the distal portions of ileum, see line 205

  • Specify whether or not the digesta was flushed out from the intestinal sections before fixing in formalin

Response: Thanks a lot, Thanks alot, yes flushed out from the intestinal sections; see line 206

  • Be consistent with either P<0.05 or P≤0.05

Response: Thank you very much. It was revised to be consistent as kindly recommended (p<0.05)

  • "Tukey post-hoc test" not Tukey post-hock test

Response: Thanks a lot. Done as kindly recommended, Line 222

7) Decisions on improved gut microbiota were made based on only Lactobacillus and Coliform bacteria, so I suggest changing the more specific title.

Response: Thanks a lot. Done as kindly recommended; see the title lines 2-5.

Reviewer 3 Report

Comments and Suggestions for Authors

The work is very interesting and brings new elements regarding the use of Spirulina platensis and/or Allium Sativum on the antioxidant status, immune response, intestinal morphology and intestinal microorganisms of broiler chickens subjected to heat stress.

Based on well-selected literature, the authors justified the purposefulness of the research conducted. The aim of the work was clearly formulated. The material used for research is sufficient, the research methods have been selected appropriately and described very precisely. The results were presented in 7 tables, graphs and photographs and discussed in detail.

The conclusions are correct and result from the obtained research results.

Comments on the Quality of English Language

Author Response

Reviewer 3

The work is very interesting and brings new elements regarding the use of Spirulina platensis and/or Allium Sativum on the antioxidant status, immune response, intestinal morphology and intestinal microorganisms of broiler chickens subjected to heat stress.

Based on well-selected literature, the authors justified the purposefulness of the research conducted. The aim of the work was clearly formulated. The material used for research is sufficient, the research methods have been selected appropriately and described very precisely. The results were presented in 7 tables, graphs and photographs and discussed in detail.

The conclusions are correct and result from the obtained research results.

Thank you very much for this positive feedback.

Reviewer 4 Report

Comments and Suggestions for Authors

I read the manuscript entitled " Effects of Spirulina platensis and/or Allium Sativum on Antioxidant Status, Immune Response, Gut Morphology, and Intestinal Microbial of Heat-Stressed Broiler Chicken." with great interest. The methods are adequately described, but there are many issues to resolve.

Lines 69–71: provide refs

Lines 80-82: Mention the potential role of spirulina supplementation (cite doi:10.1080/09637486.2022.2137785 and doi: 10.3389/fnut.2022.1048258).

lines 54-58. Add some refs.

Line 396-397: add ref

Line 309-403: add ref.

In general, the discussion needs to be clear about what the practical question is that you are trying to address. How is the answer to this question important to the field? How is this study impactful and not trivial? This needs more clarity as well.

Author Response

I read the manuscript entitled " Effects of Spirulina platensis and/or Allium Sativum on Antioxidant Status, Immune Response, Gut Morphology, and Intestinal Microbial of Heat-Stressed Broiler Chicken." with great interest. The methods are adequately described, but there are many issues to resolve.

Lines 69–71: provide refs

Response: Thank you very much. A reference was added as kindly recommended, The diverse pharmacological and nutritional benefits of algae (Spirulina spp.) and garlic (Allium sativum) [14,15], Line 85.

Lines 80-82: Mention the potential role of spirulina supplementation (cite doi:10.1080/09637486.2022.2137785 and doi: 10.3389/fnut.2022.1048258).

Response: Thank you very much. These references were added as kindly recommended. Additionally, SP has high concentrations of bioactive substances, such as phycocyanins, steroids, saponins, chlorophyll, carotene, flavonoids, triterpenoids, and phenolic acids, which exhibit wide biological effects, including antibacterial, antiviral, anti-inflammatory, hepatoprotective, and antioxidant characters [17–21].Line 94.

lines 54-58. Add some refs.

Response: Thanks a lot. Chronic stress, known as the “secret killer” in poultry, results from commercial chicken farming, intestinal microbiota imbalance, mycotoxins, diet-mediated oxidative stress, environmental stressors, and nutritional imbalances [1–3], Lines 64-67

Line 396-397: add ref, Line 309-403: add ref.

Response: Thanks a lot, these two sentences were deleted

In general, the discussion needs to be clear about what the practical question is that you are trying to address. How is the answer to this question important to the field? How is this study impactful and not trivial? This needs more clarity as well.

Thank you very much. The discussion was re-written to be more clear, Lines 361-469

Round 2

Reviewer 2 Report

Comments and Suggestions for Authors

The author improved the manuscript significantly. However, I still have some concerns and recommendations, which you can access in the attached PDF file.

Author Response

Thank you very much again for your time. All raised comments that vimproved our manuscript have been addressed. 

1) spp. was revised throughout the manuscript to be not italics. Line 77, 399, 404, 408)

2) The antioxidant capacity of SP was determined according to Prieto et al. (Line 121)

3) ...unsexed ROSS-308 broiler chicks (Line 130, Table 6

4) Table 8. The ME is the same in both diets according to NRC 1994. Nutrition guide for broilers (Ref no. 30) during the 0-6 weeks of age

5) Digesta was flushed out using PBS, line 197

6) Suggested references have been cited, Lines 242-243

7) Future prespectives have been added to the conclusion (Lines 435-438)

8) The potential preboiotic effect of SP has been discussed (Line 401)

9)  New references have been cited in the Reference List, Line 544- 562 and 616-618

Reviewer 4 Report

Comments and Suggestions for Authors

well done

Author Response

Thank you very much for time and comments that improved our manuscript